# Assessment of Apical Patency in Permanent First Molars Using Deep Learning on CBCT-Derived Pseudopanoramic Images: A Retrospective Study

**DOI:** 10.3390/bioengineering12111233

**Published:** 2025-11-11

**Authors:** Suna Deniz Bostanci, Zeliha Hatipoğlu Palaz, Kevser Özdem Karaca, Muhammet Ali Akcayol, Mehmet Bani

**Affiliations:** 1Private Practice, 59860 Tekirdağ, Turkey; sudebostanci@gmail.com; 2Department of Pediatric Dentistry, Faculty of Dentistry, Gazi University, 06490 Ankara, Turkey; mehmetbani@gazi.edu.tr; 3Department of Computer Engineering, Faculty of Engineering, Gazi University, 06490 Ankara, Turkey; kevserozdem@gazi.edu.tr (K.Ö.K.); akcayol@gazi.edu.tr (M.A.A.)

**Keywords:** convolutional neural networks (CNNs), dental imaging, dentistry, apical closure detection, pseudopanoramic radiographs, artificial intelligence in dentistry

## Abstract

**Background:** Assessment of root development and apical closure is critical in dental disciplines, including endodontics, trauma management, and age estimation. This study aims to leverage advances in deep learning Convolutional Neural Networks (CNNs) to automatically evaluate the apical region status of permanent first molars, highlighting a digital health application of AI in dentistry. **Methods:** In this retrospective study, 262 Cone Beam Computed Tomography (CBCT) scans were reviewed, and 147 anonymized dental images were cropped from pseudopanoramic radiographs, including standard measurements. Tooth regions were resized to 471 × 1075 pixels and split into training (80%) and test (20%) sets. CNN performance was assessed using accuracy, precision, recall, F1-score, and receiver operating characteristic (ROC) curves with area under the curve (AUC), demonstrating AI-based image analysis in a dental context. **Results:** Precision, recall, and F1-scores were 0.79 for open roots and 0.81 for closed roots, with a macro average of 0.80 across all metrics. The overall accuracy and AUC were also 0.80. **Conclusions:** These results suggest that CNNs can be effectively used to assess apical patency from ROI images derived from pseudopanoramic radiographs.

## 1. Introduction

The stages of apex formation, tooth eruption, and apical closure are critical developmental processes that have been extensively studied in various contexts, including regenerative endodontic treatment, trauma management, and age estimation [1]. Several methods have been developed to evaluate these stages, such as those proposed by Moorrees, Nolla, Cvek, and Demirjian [2,3,4]. The Nolla method classifies tooth development into ten stages, assessing the teeth on one side of both the maxilla and mandible; third molars are excluded from this assessment. Age estimation is conducted using sex-specific tables for boys and girls [2]. In the Demirjian method, the development of seven permanent mandibular left teeth is evaluated in eight stages based on the width of the pulp chamber and the degree of calcification [2]. Moorrees categorized the stages of tooth development by separately assessing crown formation, root development, and apical closure, with the latter recorded in two distinct stages following root completion. Cvek proposed a five-stage classification, wherein an apical opening of 1 mm or less is considered indicative of apical closure [3,4]. Permanent first molars typically erupt between the ages of 6 and 7 [5]. Root development and apical closure are usually completed approximately three years after eruption [1]. Enamel maturation continues even after eruption, a process referred to as post-eruption maturation [5,6]. During this period, the risk of complications such as trauma, caries due to inadequate oral hygiene, or infections arising from developmental anomalies like dens invaginatus increases. Without timely intervention, trauma may result in loss of pulp vitality or necessitate root canal treatment due to secondary infections [1,7]. In permanent first molars, the stage of root development—particularly the status of the apical foramen—is a key determinant in selecting an appropriate treatment strategy. Clinical and radiographic evaluations guide clinicians in determining whether to proceed with root canal treatment, apexification, or regenerative endodontic therapy. Among these, regenerative procedures are primarily dependent on the apical diameter, as immature teeth with wider apical openings and a higher concentration of apical stem cells are considered more favorable for pulp regeneration [8]. In pediatric patients, establishing a well-defined treatment plan from the outset is crucial for maintaining cooperation and minimizing the need for treatment modifications during the clinical process [9]. Therefore, during radiographic assessments, it is essential to determine whether root development has been completed. This evaluation may be conducted using panoramic radiographs, periapical radiographs, or cone-beam computed tomography (CBCT). CBCT provides high-resolution three-dimensional imaging and is considered the most accurate method; however, it is not always accessible and subjects patients to relatively higher radiation doses. As a result, periapical and panoramic radiographs remain the most widely used techniques. Nonetheless, panoramic radiographs offer only two-dimensional images, which may suffer from distortion, limited resolution, and superimposition of anatomical structures—especially in the posterior region—making interpretation more challenging [10].

In recent years, artificial intelligence (AI) has gained significant prominence in medical and dental research due to its ability to perform diagnostic and predictive tasks that traditionally require human expertise [11]. AI systems, particularly convolutional neural networks (CNNs), have been successfully applied in various dental specialties. In endodontics, they have been used to detect periapical lesions, identify root fractures, and evaluate canal morphology. In prosthodontics, AI facilitates the design and fabrication of crowns and dentures by integrating with CAD/CAM and 3D printing systems, thereby enhancing precision and efficiency. In oral and maxillofacial radiology, AI assists in the detection of caries, cysts, and anatomical structures with high accuracy [12,13,14,15,16]. Moreover, AI-based educational platforms are emerging as practical tools for improving clinical training and decision-making skills among dental students and practitioners [17].

Recent reviews have highlighted the increasing integration of AI—particularly deep learning and large language models—into dental diagnostics, treatment planning, and education. For instance, ChatGPT-3.5 and ChatGPT-4 (OpenAI, San Francisco, CA, USA) and similar models have demonstrated potential as accessible tools to support clinical reasoning, enhance communication, and facilitate learning in dental education and practice [18]. These advancements underscore a broader shift toward digital and AI-driven dentistry, where automated image interpretation and intelligent systems can complement traditional diagnostic workflows.

Therefore, the present study hypothesizes that convolutional neural network (CNN)-based deep learning models can accurately and reliably determine the apical closure status of permanent first molars on CBCT-derived pseudopanoramic radiographs, providing an effective digital tool to assist clinicians in assessing root development.

## 2. Materials and Methods

Ethical approval for this study was obtained from the Gazi University Ethics Commission (approval code: 2023-1408; date of approval: 21 November 2023). This retrospective study evaluated CBCT images of pediatric patients aged between 5 and 12 years.

Inclusion Criteria:Radiographs with sufficient image quality for diagnostic evaluation,Precise assessment of whether the root apex is open or closed on CBCT images,Absence of any pathology adjacent to or associated with the roots of the tooth to be evaluated,No prior treatments such as regenerative endodontic therapy or apexification were applied to the tooth in question,The tooth being evaluated is in the post-eruptive stage.

Exclusion Criteria:Radiographs with insufficient image quality for diagnosis,Inability to clearly determine the status (open or closed) of the root apex on CBCT images,Presence of pathology adjacent to or involving the roots of the tooth to be evaluated,History of treatments such as regenerative endodontic therapy or apexification on the tooth to be assessed,The tooth is in the pre-eruptive stage,Presence of a dental anomaly in the tooth to be evaluated (e.g., dentinogenesis imperfecta, taurodontism).

### 2.1. Sample Size and Power Analysis

In this study, the sample size was calculated using a single-proportion power analysis with a 95% confidence level and 80% statistical power (β = 0.20). The total accessible population consisted of 262 cases. The expected proportion (p) of the target outcome was assumed to be 0.50, providing the maximum sample size under a conservative assumption. Considering a significance level of α = 0.05, the corresponding test statistic (Z) was 1.96. Based on these parameters, the minimum required sample size was calculated as 156 images. The primary endpoint of the study was to evaluate the performance of the CNN model in correctly classifying the apical status (open or closed) of permanent first molars.

CBCT images were taken as the gold standard. CBCT images were obtained from the DentisTomo-Dental Imaging Center archive. In the device used, irradiations were performed with the HDX WILL Dentri 3D (HDX WILL Corp., Seoul, Republic of Korea) CBCT device in a 160 × 80 mm field of view (FOV) area. Dose adjustments were made according to the patient’s body type by a single technician. Images were created with a voxel size of 200 µm. All measurements used in the study were recorded by a single examiner on axial, sagittal, coronal, and cross-sectional slices using Cybermed OnDemand software version 1.0.10.6388 (Cybermed Inc., Daejeon, Republic of Korea). Pseudopanoramic radiographs were generated from CBCT scans.

For standardization, the cervical region of the lower anterior teeth was used as a reference point. The imaging area was taken with a thickness of 15 mm. The resulting pseudopanoramic radiographs were recorded. A numbering system was used, without mentioning the patient’s name, in accordance with the commitment under the Personal Data Protection Law. Cropped images obtained from these radiographs were included in the study. In the CBCT images, apical closure was assessed in the axial section. According to the Cvek classification, teeth with an apical opening of 1.0 mm or less were considered to have a closed apex. In contrast, those with larger openings were classified as having an open apex [3]. Regions of interest (ROIs) were extracted from the relevant areas, with dimensions ranging from 471 to 1075 pixels, including only the root portion of the permanent first molar teeth, using ImageJ software version 1.53 (Figure 1a,b). For the study, cropped images were taken from both open and closed root apex views. ImageJ was used for this purpose. To ensure that the cropped images were standardized, the images were scanned to a 471 × 1075 image size, ensuring that all images contained the anatomical structures planned for analysis. After this, sections were extracted from all pseudopanoramic images using ImageJ at these dimensions.

### 2.2. Creating the Model

A deep learning model based on a Convolutional Neural Network (CNN) architecture was developed to classify images into two categories: open and closed apices. CNN architectures have been widely used in medical and dental imaging due to their proven success in recognizing spatial hierarchies and feature extraction from image data.

### 2.3. Model Architecture

The model architecture consisted of three convolutional layers and three max-pooling layers arranged sequentially, followed by fully connected (dense) layers for classification.

In each convolutional layer, feature maps were generated by applying convolution filters to the outputs of the preceding layer. The convolution layers employed 3 × 3 kernels with increasing filter sizes of 32, 64, and 128, allowing the model to extract both low- and high-level features progressively. All convolutional layers utilized the Rectified Linear Unit (ReLU) activation function, defined in Equation (1):(ReLU(x) = max (0,x)(1) ReLU was preferred due to its computational efficiency and capability to mitigate vanishing gradient problems during backpropagation.

### 2.4. Pooling and Regularization

After each convolutional layer, a MaxPooling2D layer with a pool size of 2 × 2 was used to reduce the spatial dimensions, thereby minimizing overfitting and computational cost while preserving the most salient features. To further prevent overfitting, a Dropout layer with a rate of 0.3 was incorporated before the fully connected layers, ensuring model generalization by randomly turning off 30% of neurons during training.

### 2.5. Classification Layers

The extracted features from the final pooling layer were flattened and fed into dense layers for classification. The final output layer employed a Softmax activation function, enabling the model to output class probabilities for the two categories (open and closed). Equation (2) represents the SoftMax function:(2)$$S(x)=\frac{e^{x_{i}}}{\sum_{j=1}^{n}e^{x_{j}}}$$

### 2.6. Dataset Preparation and Augmentation

A total of 147 CBCT-derived pseudopanoramic images were used, comprising 67 open and 80 closed apex samples. The dataset was split into training (70%), validation (10%), and testing (20%) subsets. During this process, the class ratio (open/closed) was preserved across all subsets using a stratified random sampling approach.

To enhance dataset diversity and mitigate class imbalance, the training data underwent data augmentation by applying vertical flipping, which increased the number of training samples from 117 to 217 images. Augmentation not only improved the model’s ability to generalize to unseen data but also minimized the risk of overfitting to the limited dataset.

### 2.7. Implementation Details

The study has been conducted using a computer having a RAM of 16 GB, Intel Core i7–13700HX as CPU and NVIDIA GeForce RTX4060 as GPU. The model was implemented using Python (version 3.10) as the programming language, with the help of TensorFlow 2.16 and Keras 3.0 deep learning frameworks. The Adam optimizer was employed with an initial learning rate of 0.001, and the categorical cross-entropy loss function was used for optimization. Model performance was monitored via validation accuracy and loss across epochs.

### 2.8. Model Evaluation

The performance of the final trained model was assessed using accuracy, precision, recall, and F1-score metrics on the test set. A five-fold cross-validation approach has been used to obtain more reliable results. The dataset is divided into five parts called folds. The training process is conducted on four of the folds and the remaining fold is used for testing. This process is repeated five times, using different folds for testing. The performance of the model is the average of the results obtained from these five iterations. A confusion matrix has been generated to visualize the classification performance. The architecture’s flow diagram is illustrated in Figure 2, and the detailed layer configuration, including output dimensions and parameter counts, is provided in Table 1.

### 2.9. Statistical Analysis

The study focused on evaluating the performance of a deep learning-based classification model. The model’s diagnostic capability was assessed using accuracy, precision, recall, F1-score, and the area under the receiver operating characteristic (ROC) curve (AUC), which are standard statistical performance metrics used in artificial intelligence-based diagnostic studies. Where applicable, a significance threshold of *p* < 0.05 was adopted, and 95% confidence intervals (CIs) were calculated to ensure analytical rigor and reproducibility.

## 3. Results

In the study results, 262 tomographies were examined, and 147 CBCTs were included. Of the CBCTs included, 54% (80) were found to have closed root apices, while 46% (67) had open root apices. The actual and predicted labels are presented using a confusion matrix in Table 2.

According to the confusion matrix obtained from the predictions made on the test data, 11 out of 14 teeth with open apices correspond to the real values, while 13 out of 16 teeth with closed apices match the real values.

CNN accuracy, precision, recall, F1 score, and receiver operating characteristic (ROC) curve and area under the curve (AUC) were evaluated. As a result of evaluating the data with CNN, the precision, recall, and F1 score values of teeth with open roots were found to be 0.79. The precision, recall, and F1 score value of teeth with closed root ends were found to be 0.81. The macro average values of precision, recall, and F1 score are 0.80. The accuracy of apical evaluations was calculated as 0.80. The AUC value was calculated to be 0.80. The results are given in Table 3 and Table 4.

## 4. Discussion

Root development is typically completed, and the root apex closes approximately three years after tooth eruption. During this critical period, if a pulpal procedure is required, the status of the root apex—whether open or closed—serves as a key determinant in selecting the appropriate treatment approach. Various radiographic methods are employed to assess the stage and monitor the progress of root development [9].

With the integration of artificial intelligence (AI) into healthcare, convolutional neural network (CNN) algorithms have been developed to support clinicians in diagnostic and treatment planning processes. In endodontics, CNN-based models have been applied in areas such as analyzing root canal anatomy, estimating canal length, detecting periapical lesions and root fractures, and predicting the outcomes of root canal retreatment procedures [19].

In pediatric dentistry, determining apical closure is particularly important for planning pulpal interventions. In this study, CNN algorithms were utilized to classify whether the root apex of permanent first molars was open or closed. Cone-beam computed tomography (CBCT) images served as the reference standard for determining apical status, and the accuracy of CNN-based classifications was evaluated accordingly. Our model achieved an overall accuracy of 0.80, with sensitivity of 0.79 for open apices and 0.81 for closed apices, indicating balanced class performance. These metrics imply that the CNN learned discriminative features related to apical maturation even when working with two-dimensional pseudopanoramic projections.

Regions of interest (ROIs) were labeled as either “open” or “closed” root apices. To ensure consistency and reduce variability, only CBCT scans obtained by a single operator were included. The ROIs used in the analysis were extracted from standardized pseudopanoramic images, with dimensions ranging from 471 to 1075 pixels. This standardization likely contributed to reduced technical variance, supporting more reliable learning and evaluation.

The sensitivity of the model was calculated to be 0.79 for teeth with open apices and 0.81 for those with closed apices. The area under the curve (AUC) value was determined to be 0.79. In a related study, Patel et al. investigated the detection of periapical lesions using both periapical radiographs and CBCT, with CBCT serving as the gold standard. They found that only 24.8% of lesions were detectable on periapical radiographs, with detection accuracy increasing for larger lesions. This finding highlights the importance of root developmental stage in diagnostic imaging. The overall sensitivity for lesion detection via periapical radiographs was 0.24, and the corresponding AUC was reported as 0.791 [20]. Taken together, whereas two-dimensional radiographs showed limited sensitivity in Patel et al., our results suggest that a CNN can partially compensate for the inherent limitations of two-dimensional modalities by extracting subtle textural and shape-based features—thereby improving practical utility when CBCT is unavailable or unwarranted.

Saghiri et al. employed an artificial neural network to determine the location of the minor apical foramen using 50 single-rooted teeth extracted for periapical reasons [21]. The minor apical foramen was radiographically identified in cadaver specimens using K-files, and the teeth were examined under a stereomicroscope as the gold standard. The apical foramen was correctly identified in 93% of the cases. In another similar study, the same research group compared radiographic evaluations between endodontists and artificial neural networks using cadaver samples. The apical foramen was correctly identified by endodontists with an accuracy of 0.76, whereas the artificial neural networks achieved a higher accuracy of 0.96 [22].

In contrast, our study focused not on the localization of the apical foramen, but on evaluating the apical patency status. In teeth with incomplete root development, the apical foramen was not yet formed, whereas in teeth with completed root development, apical closure was assessed. The apical status was correctly determined with an accuracy of 0.80. Thus, while the studies by Saghiri et al. primarily investigated anatomical localization, our research evaluated a developmental process. Both approaches, however, support the concept that machine learning can match or even surpass human performance in diagnostic tasks related to the apical region.

Setzer et al. detected periapical lesions by examining CBCT images in the sagittal, axial, and coronal planes. The images were analyzed using a deep learning approach, and a total of 61 images were included in their study. Lesions were correctly identified with an accuracy rate of 0.93 [23].

In our study, the images were examined in the axial section using CBCT software version 1.0 (Build 1.0.10.4490). However, the regions of interest (ROIs) were not directly extracted from CBCT sections, but rather obtained from standardized pseudopanoramic radiographs generated within the software. Although some studies have accepted three-dimensional images as the gold standard [20], CBCT scans were used as the reference standard in our study to ensure consistency and accuracy. The ROIs used to form the training and testing sets were extracted from pseudopanoramic radiographs, which are considered a two-dimensional imaging modality. Accordingly, although deep learning analyses based on three-dimensional CBCT sections can achieve high accuracy, our approach provides a more accessible and lower-radiation alternative for assessing apical closure using standardized two-dimensional outputs derived from CBCT workflows.

To the best of our knowledge, there are only a limited number of studies [1,24] that have directly focused on CBCT-based evaluations of apical closure or patency, which highlights the novelty and relevance of the present study. Previous studies have focused on periapical lesions and the apical foramen; however, none have specifically assessed apical patency using CBCT. These studies were used to contextualize and support the findings derived from our dataset. In our study, the absence of a visible apical foramen in teeth with open apices, as well as the lack of similar CBCT-based evaluations of apical patency, represent important limitations. Moreover, the relatively small dataset size and the use of images obtained from a single imaging center may limit the generalizability of the results and introduce potential selection or institutional bias. Another limitation is the use of only the CNN algorithm for evaluating apical patency, without comparison to other architectures. However, our findings may contribute to and support future studies in this field. In fact, it would be interesting to match CNNs with both smartphone applications [25] and other artificial intelligence-guided software [26] in order to improve data and potential applications in daily clinical practice.

This innovation positions our work as a foundational step toward the clinical implementation of automated assessment of apex status in pediatric dentistry. Accordingly, future research may include multi-center prospective datasets, comparing different deep learning architectures (such as ensemble CNNs or transformer-based models), performing external validation studies, and evaluating the integration of developed models into clinical workflows. In clinical practice, an accuracy of 0.80 may help clinicians identify apical status more consistently, supporting timely decision-making in pediatric endodontic procedures where maintaining pulp vitality and minimizing overtreatment are crucial. From a clinical perspective, the achieved accuracy of 0.80 is promising but should be interpreted with caution. While this level of performance suggests that CNN-based models can reliably assist in the assessment of apical status, clinical decision-making in pediatric endodontics requires high precision, and AI systems should be considered as supportive tools rather than replacements for expert evaluation. In conclusion, our findings demonstrate that CNN-based models can accurately assess apical closure from standardized pseudopanoramic images. This approach has the potential to bridge the gap between high-resolution three-dimensional imaging and routinely used two-dimensional modalities, thereby supporting faster, more accessible, and better-informed treatment planning in pediatric dentistry.

## 5. Conclusions

Accurate evaluation of root apices on radiographic images is essential in pediatric dentistry. Previous CNN-based studies in this field have primarily focused on teeth with completed root maturation. In contrast, our study assessed apical patency and achieved an accuracy rate of 0.80. In recent years, there has been a rapid increase in research involving artificial intelligence in healthcare. As no similar studies evaluating apical patency using CBCT and CNN have been reported to date, our study provides a foundational step for future research in this area.

## Figures and Tables

**Figure 1 bioengineering-12-01233-f001:**
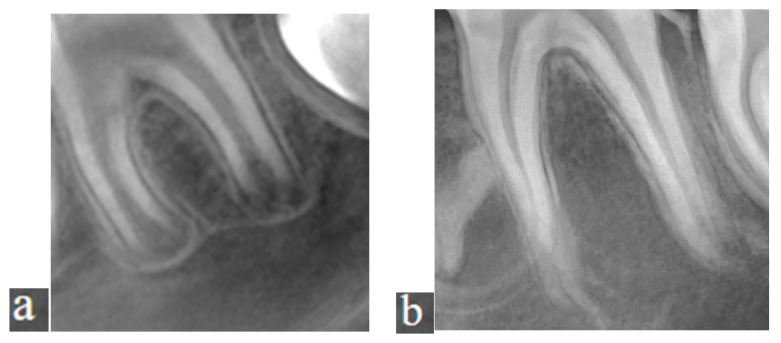
(**a**) The ROI of tooth number 36 with an open apex, taken from the pseudopanoramic radiograph; (**b**) The ROI of tooth number 36 with a closed apex, taken from the pseudopanoramic radiograph.

**Figure 2 bioengineering-12-01233-f002:**
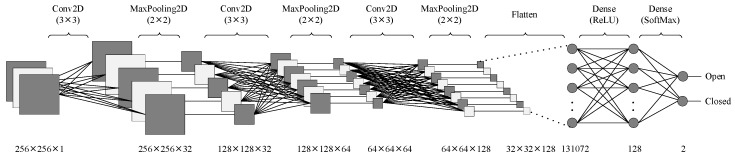
Layers of the developed model.

**Table 1 bioengineering-12-01233-t001:** Parameters of the Developed Model.

Layer	Output Shapes	Parameter Numbe
Conv2D_1	256 × 256 × 32	320
MaxPooling2D_1	128 × 128 × 32	0
Conv2D_2	128 × 128 × 64	18.496
MaxPooling2D_2	64 × 64 × 64	0
Conv2D_3	64 × 64 × 128	73.856
MaxPooling2D_3	32 × 32 × 128	0
Flatten_1	131,072	0
Dense_1	128	16.777.344
Dropout_1	128	0
Dense_2	2	258
Toplam parametre sayısı		16.870.274

**Table 2 bioengineering-12-01233-t002:** Confusion Matrix.

		Predicted	
		Open	Closed
Actual	Open	11	3
	Closed	3	13

**Table 3 bioengineering-12-01233-t003:** CNN performance metrics for open and closed root apices.

	Precision	Recall	F1-Score	Support
Open	0.79	0.79	0.79	14
Close	0.81	0.81	0.81	16
Macro avg	0.80	0.80	0.80	30
Weighted avg	0.80	0.80	0.80	30

**Table 4 bioengineering-12-01233-t004:** The accuracy and AUC values.

	Accuracy	AUC
Value	0.80	0.80

## Data Availability

The data supporting the results of this study may be available from the corresponding author upon request due to privacy and ethical restrictions.

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
