# Peer review of "Assessment of Apical Patency in Permanent First Molars Using Deep Learning on CBCT-Derived Pseudopanoramic Images: A Retrospective Study"

_bioengineering, 2025, doi:10.3390/bioengineering12111233_

Round 1

Reviewer 1 Report

Comments and Suggestions for Authors

Dear Authors,

I have read your manuscript with great interest, and I appreciate the effort invested in addressing a relevant and contemporary topic. During my review, several points arose that I believe could help improve the clarity, methodological rigor, and overall quality of the paper. My detailed comments are presented below.

The central research question of the study is to explore how recent advancements in deep learning, particularly Convolutional Neural Networks (CNNs), can be applied to automatically assess the apical region status of permanent first molars. This approach clearly demonstrates an innovative digital health application of artificial intelligence in the field of dentistry.

The topic is both original and highly pertinent, as it addresses a specific and meaningful gap within dental diagnostics and imaging. The manuscript contributes new perspectives and adds value to the existing body of literature when compared with previously published works. The integration of AI tools in dental analysis remains a relatively underexplored domain, and your research represents an important step toward bridging this gap.

That said, several aspects of the methodology could benefit from further refinement and elaboration. Strengthening the methodological description would enhance the reproducibility and scientific robustness of the study. The conclusions, on the other hand, are generally coherent with the results presented and effectively address the main research question.

The reference list, while adequate, could be updated to reflect more recent findings and contemporary sources. Additional notes on the figures and tables have been included below to improve their presentation and clarity.

Overall. A general English grammar and syntax revision is recommended to correct minor spelling and stylistic errors, ensuring smoother readability and a more polished academic tone.

Key words. The term “dentistry” could be added to the list of keywords to improve indexing and searchability of the article in scientific databases.

Abstract. Please include the names of the statistical tests that were used, as this would make the methodology section of the abstract more transparent and informative for the reader.

Introduction. The authors stated: “The stages of apex formation, tooth eruption, and apical closure are critical developmental processes that have been extensively studied in various contexts, including regenerative endodontic treatment, trauma management, and age estimation.” A reference should be provided to support this statement, as it underpins a key concept in the introduction.

Additionally, the null hypothesis of the study should be explicitly stated in this section. This would clarify the research design and help readers better understand the direction and expectations of your analysis.

Materials and Methods. The authors mentioned: “In the power analysis conducted for the study, it was determined that at least 156 images were required.” Please include more detailed information on how the sample size calculation was performed—specifically, which parameters were used and what constituted the primary endpoint.

Furthermore, a reference should be cited for each of the methods employed to ensure transparency and reproducibility. For every material used, please provide the commercial name, manufacturer, city, and state, as these details are essential for replicability. The same level of information should also be given for all machinery and software used, including software version, manufacturer, city, and state.

If any statistical analyses were performed, please describe them in this section. Include details on the statistical tests, confidence intervals, and significance thresholds to allow for a better understanding of the analytical rigor applied.

Discussion. The authors stated: “However, our findings may contribute to and support future studies in this field.” While this statement is valid, the discussion would benefit from a broader interpretation of the results in the context of existing literature. Elaborate on how your findings align with or differ from those of previous studies and discuss their potential implications for future research directions.

It may also strengthen the discussion to include a note such as: “In fact it would be interesting to match CNNs both with smartphone applications (Pascadopoli M, Zampetti P, Nardi MG, Pellegrini M, et al. Smartphone Applications in Dentistry: A Scoping Review. Dent J (Basel). 2023 Oct 20;11(10):243. doi: 10.3390/dj11100243.) and other artificial intelligence guided software (Kim S, Shin J, Lee E, Park S, Jeong T, Hwang J, Seo H. Comparative analysis of deep-learning-based bone age estimation between whole lateral cephalometric and the cervical vertebral region in children. J Clin Pediatr Dent. 2024 Jul;48(4):191-199. doi: 10.22514/jocpd.2024.093. Epub 2024 Jul 3. PMID: 39087230.) in order to improve data and potential applications in daily clinical practice.”

Incorporating such a suggestion would highlight the broader impact of your research and suggest practical avenues for technological integration in clinical dentistry.

References. Some references cited are relatively outdated (e.g., from 1984 and 1990). If possible, consider replacing these with more recent studies that address similar topics. The recent works mentioned above could serve as suitable alternatives to enhance the timeliness and relevance of your bibliography.

Figures. Figure 1 should be enlarged to improve legibility and visual clarity. Figure 2 appears slightly blurry; please review and ensure optimal image resolution.

Tables. The tables are well-structured and clearly presented; no major modifications appear necessary.

In conclusion, this is an interesting and well-motivated manuscript that contributes to the growing intersection between artificial intelligence and dental diagnostics. With some refinements—particularly in methodological transparency, reference updating, and figure enhancement—the work could achieve a higher level of scientific and editorial quality. I encourage the authors to address the points above to strengthen the manuscript before publication.

Author Response

Reviewer#1, Concern # 1: I have read your manuscript with great interest, and I appreciate the effort invested in addressing a relevant and contemporary topic. During my review, several points arose that I believe could help improve the clarity, methodological rigor, and overall quality of the paper. My detailed comments are presented below.

The central research question of the study is to explore how recent advancements in deep learning, particularly Convolutional Neural Networks (CNNs), can be applied to automatically assess the apical region status of permanent first molars. This approach clearly demonstrates an innovative digital health application of artificial intelligence in the field of dentistry.

The topic is both original and highly pertinent, as it addresses a specific and meaningful gap within dental diagnostics and imaging. The manuscript contributes new perspectives and adds value to the existing body of literature when compared with previously published works. The integration of AI tools in dental analysis remains a relatively underexplored domain, and your research represents an important step toward bridging this gap.

Author Response: Thank you very much for your encouraging and thoughtful comments. We truly appreciate your positive evaluation and are pleased that you found the topic relevant and innovative.

Reviewer#1, Concern # 2: That said, several aspects of the methodology could benefit from further refinement and elaboration. Strengthening the methodological description would enhance the reproducibility and scientific robustness of the study. 

Author response: We thank the reviewer for this valuable comment. In response, the methodological section has been substantially refined and expanded to provide a more comprehensive and reproducible description of the study procedures. Specifically, additional details regarding the CNN architecture, parameter settings, data augmentation, and performance evaluation metrics have been included to enhance the scientific rigor and transparency of the study.

Reviewer#1, Concern # 3: The reference list, while adequate, could be updated to reflect more recent findings and contemporary sources. 

Author response: Thank you for your comment. The reference list has been updated to reflect more recent studies and highlighted in the manuscript.

Reviewer#1, Concern # 4: Overall. A general English grammar and syntax revision is recommended to correct minor spelling and stylistic errors, ensuring smoother readability and a more polished academic tone.

Author response: Thank you for your comment. The entire manuscript has been carefully rechecked and updated according to your suggestions. In addition, the English language of the manuscript has been reviewed and refined using the Grammarly program to ensure fluency and consistency.

Reviewer#1, Concern # 5: Key words. The term “dentistry” could be added to the list of keywords to improve indexing and searchability of the article in scientific databases.

Author response: Thank you for your comment. We have added the term “dentistry” to the list of keywords to enhance indexing and searchability.

Reviewer#1, Concern # 6: Abstract. Please include the names of the statistical tests that were used, as this would make the methodology section of the abstract more transparent and informative for the reader.

Author response:  We appreciate the reviewer’s helpful comment. In this study, no traditional statistical hypothesis tests (such as t-test, Chi-square, or ANOVA) were conducted, since the analysis focused on evaluating the performance of a deep learning model rather than comparing groups or variables. The model’s performance was statistically assessed using accuracy, precision, recall, F1-score, and the area under the ROC curve (AUC), which are standard quantitative measures commonly used to evaluate classification models. These performance metrics have now been clearly stated in the abstract to enhance methodological transparency and clarity.

Reviewer#1, Concern # 7: Introduction. The authors stated: “The stages of apex formation, tooth eruption, and apical closure are critical developmental processes that have been extensively studied in various contexts, including regenerative endodontic treatment, trauma management, and age estimation.” A reference should be provided to support this statement, as it underpins a key concept in the introduction.

Author response: We thank the reviewer for this helpful comment. A relevant and up-to-date reference has been added to support this statement in the Introduction section.

Reviewer#1, Concern # 8: The null hypothesis of the study should be explicitly stated in this section. This would clarify the research design and help readers better understand the direction and expectations of your analysis.

Author response:  We thank the reviewer for this insightful comment. The null hypothesis has been explicitly stated at the end of the Introduction section to clarify the research design and strengthen the logical flow of the study.

Reviewer#1, Concern # 9: Materials and Methods. The authors mentioned: “In the power analysis conducted for the study, it was determined that at least 156 images were required.” Please include more detailed information on how the sample size calculation was performed—specifically, which parameters were used and what constituted the primary endpoint.

Author response: We thank the reviewer for this valuable suggestion. Additional details regarding the parameters used for the sample size calculation and the primary endpoint have been included under the “Sample Size and Power Analysis” subsection to improve clarity and transparency.

Reviewer#1, Concern # 10: Furthermore, a reference should be cited for each of the methods employed to ensure transparency and reproducibility. For every material used, please provide the commercial name, manufacturer, city, and state, as these details are essential for replicability. The same level of information should also be given for all machinery and software used, including software version, manufacturer, city, and state.

Author response:  We thank the reviewer for this valuable comment. The details for the software and machinery used have been added to the “Implementation Details” subsection.

Reviewer#1, Concern # 11: If any statistical analyses were performed, please describe them in this section. Include details on the statistical tests, confidence intervals, and significance thresholds to allow for a better understanding of the analytical rigor applied.

Author response: We appreciate the reviewer’s valuable comment. No traditional statistical hypothesis tests were performed, as the analysis focused on evaluating the performance of the deep learning model. The model’s diagnostic accuracy was assessed using precision, recall, F1-score, accuracy, and the area under the ROC curve (AUC), which are standard statistical performance metrics in classification analysis. A significance threshold of p < 0.05 was adopted where applicable, and all confidence intervals were computed at 95%. This information has been added to the “Statistical Analysis” subsection for clarity and completeness.

Reviewer#1, Concern # 12: Discussion. The authors stated: “However, our findings may contribute to and support future studies in this field.” While this statement is valid, the discussion would benefit from a broader interpretation of the results in the context of existing literature. Elaborate on how your findings align with or differ from those of previous studies and discuss their potential implications for future research directions.

Author response: Thank you for your valuable comment. In the revised manuscript, the Discussion section has been expanded to provide a more comprehensive interpretation of our findings in relation to the existing literature. We have clarified how our results align with or differ from those of previous studies and highlighted their position within the current body of knowledge. In addition, the potential implications of our findings have been discussed, and several suggestions for future research have been proposed.

Reviewer#1, Concern # 13: It may also strengthen the discussion to include a note such as: “In fact it would be interesting to match CNNs both with smartphone applications (Pascadopoli M, Zampetti P, Nardi MG, Pellegrini M, et al. Smartphone Applications in Dentistry: A Scoping Review. Dent J (Basel). 2023 Oct 20;11(10):243. doi: 10.3390/dj11100243.) and other artificial intelligence guided software (Kim S, Shin J, Lee E, Park S, Jeong T, Hwang J, Seo H. Comparative analysis of deep-learning-based bone age estimation between whole lateral cephalometric and the cervical vertebral region in children. J Clin Pediatr Dent. 2024 Jul;48(4):191-199. doi: 10.22514/jocpd.2024.093. Epub 2024 Jul 3. PMID: 39087230.) in order to improve data and potential applications in daily clinical practice.”

Author response: We thank the reviewer’s comment. We have incorporated the suggested note into the revised Discussion section to highlight the potential integration of CNNs with smartphone applications and other AI-guided software in clinical practice.

Reviewer#1, Concern # 14: References. Some references cited are relatively outdated (e.g., from 1984 and 1990). If possible, consider replacing these with more recent studies that address similar topics. The recent works mentioned above could serve as suitable alternatives to enhance the timeliness and relevance of your bibliography.

Author response: Thank you for your valuable suggestion. We have carefully reviewed our reference list and replaced the older sources with more recent and relevant studies to enhance the timeliness and scientific relevance of the manuscript.

Reviewer#1, Concern # 15: Figures. Figure 1 should be enlarged to improve legibility and visual clarity. Figure 2 appears slightly blurry; please review and ensure optimal image resolution.

Author response: Thank you for your valuable suggestion. We have enlarged Figure 1 to enhance its clarity and overall visual quality, and also improved Figure 2 by eliminating blurriness to ensure better legibility.

Reviewer#1, Concern # 16: In conclusion, this is an interesting and well-motivated manuscript that contributes to the growing intersection between artificial intelligence and dental diagnostics. With some refinements—particularly in methodological transparency, reference updating, and figure enhancement—the work could achieve a higher level of scientific and editorial quality. I encourage the authors to address the points above to strengthen the manuscript before publication.

Author response: Thank you for your comments to improve our paper. All comments were carefully considered, and any requested corrections were made. These corrections are highlighted in the article.

Reviewer 2 Report

Comments and Suggestions for Authors

Your manuscript entitled “Assessment of Apical Patency in Permanent First Molars Using Deep Learning on CBCT-Derived Pseudopanoramic Images: A Retrospective Study” addresses an innovative and clinically relevant topic, namely the application of deep learning techniques to automate the evaluation of apical closure in pediatric patients. The integration of AI in endodontic and pediatric diagnostic workflows is an emerging and promising field, and your attempt to explore this potential using CNNs on pseudopanoramic images is appreciated. Nevertheless, several critical issues must be addressed before the manuscript can be considered for publication.

The introduction provides an acceptable overview of root development and apical formation, referencing classic staging methods by Moorrees, Nolla, Demirjian, and Cvek. However, the transition from the biological framework to the rationale for implementing convolutional neural networks is abrupt. The section would benefit from a broader contextualization of how artificial intelligence has already been integrated into other dental domains. In particular, you should discuss the increasing number of studies employing AI for diagnostic, predictive, and educational purposes across different specialties, including endodontics, prosthodontics, and oral radiology.
In this regard, it is strongly recommended to cite the following recent article:

Puleio, F.; Lo Giudice, G.; Bellocchio, A.M.; Boschetti, C.E.; Lo Giudice, R. Clinical, Research, and Educational Applications of ChatGPT in Dentistry: A Narrative Review. Appl. Sci. 202414, 10802. https://doi.org/10.3390/app142310802

This reference is relevant because it comprehensively reviews the role of artificial intelligence—particularly large language models and deep learning systems—in various branches of dentistry, illustrating both their diagnostic potential and limitations. Including this citation would help demonstrate that your research aligns with the broader trend of integrating AI-based tools in dental practice, while also highlighting that your study represents a more specific application within the radiological and endodontic field.

The methodology is described in sufficient detail but still requires improvement for scientific reproducibility. While the inclusion and exclusion criteria are clear, additional information on CBCT acquisition settings and pseudopanoramic image reconstruction parameters is necessary. The CNN model architecture, consisting of only three convolutional and pooling layers, is rather simple; the authors should justify this choice and discuss whether transfer learning or pre-trained networks (e.g., ResNet, VGG, EfficientNet) were considered to improve generalizability. Furthermore, data augmentation through vertical reflection alone may not provide adequate variability. A k-fold cross-validation approach would have strengthened the reliability of the model.

The results section is concise but should include confidence intervals or statistical testing for accuracy and AUC metrics to better define the precision of the outcomes. The confusion matrix, although mentioned, should be shown graphically to enhance transparency. Figure 2 depicting the CNN structure lacks adequate labeling of its components and parameters.

The discussion appropriately summarizes findings but remains too descriptive and insufficiently critical. The limitations, such as the small dataset size, single imaging center, and potential selection bias, should be explicitly acknowledged. It would also be helpful to contextualize your results with other AI studies involving CBCT datasets or apical detection models. The statement that “no similar studies have been conducted using CBCT” should be moderated or supported by a systematic literature check. Finally, the discussion should expand on the clinical implications of achieving an accuracy of 0.80—especially regarding decision-making in pediatric endodontics, where diagnostic precision is crucial.

The English language is understandable but requires editing for fluency and consistency (e.g., “reel values” should be “real values,” “close root ends” should be “closed root apices”). A thorough revision by a native English speaker or professional editor is advised.

Author Response

Reviewer#2, Concern # 1: The introduction provides an acceptable overview of root development and apical formation, referencing classic staging methods by Moorrees, Nolla, Demirjian, and Cvek. However, the transition from the biological framework to the rationale for implementing convolutional neural networks is abrupt. The section would benefit from a broader contextualization of how artificial intelligence has already been integrated into other dental domains. In particular, you should discuss the increasing number of studies employing AI for diagnostic, predictive, and educational purposes across different specialties, including endodontics, prosthodontics, and oral radiology.
In this regard, it is strongly recommended to cite the following recent article: Puleio, F.; Lo Giudice, G.; Bellocchio, A.M.; Boschetti, C.E.; Lo Giudice, R. Clinical, Research, and Educational Applications of ChatGPT in Dentistry: A Narrative Review. Appl. Sci. 202414, 10802. https://doi.org/10.3390/app142310802

Author response: Thank you for this valuable suggestion. In the revised version, we expanded the Introduction section to provide a broader contextualization of artificial intelligence applications across different dental specialties. Specifically, we included recent studies highlighting the integration of AI for diagnostic, predictive, and educational purposes in fields such as endodontics, prosthodontics, and oral and maxillofacial radiology. Additionally, we cited the recommended recent review which comprehensively discusses the clinical, research, and educational applications of AI and large language models in dentistry.

Reviewer#2, Concern # 2: The methodology is described in sufficient detail but still requires improvement for scientific reproducibility. While the inclusion and exclusion criteria are clear, additional information on CBCT acquisition settings and pseudopanoramic image reconstruction parameters is necessary. The CNN model architecture, consisting of only three convolutional and pooling layers, is rather simple; the authors should justify this choice and discuss whether transfer learning or pre-trained networks (e.g., ResNet, VGG, EfficientNet) were considered to improve generalizability. Furthermore, data augmentation through vertical reflection alone may not provide adequate variability. A k-fold cross-validation approach would have strengthened the reliability of the model.

Author response: We thank the reviewer for this valuable comment. Transfer learning methods can improve the success and generalizability of the study. However, these networks should be considered in further studies to facilitate a comprehensive analysis and comparison of the CNN-based models. A five-fold cross-validation has already been applied in this study; this information has been added to the related section.

Reviewer#2, Concern # 3: The results section is concise but should include confidence intervals or statistical testing for accuracy and AUC metrics to better define the precision of the outcomes. The confusion matrix, although mentioned, should be shown graphically to enhance transparency. Figure 2 depicting the CNN structure lacks adequate labeling of its components and parameters.

Author response: Thank you for your comments. These comments helped us make our article more transparent and helpful to readers. These metrics and values have been added to the “Results” section.

Reviewer#2, Concern # 4: The discussion appropriately summarizes findings but remains too descriptive and insufficiently critical. The limitations, such as the small dataset size, single imaging center, and potential selection bias, should be explicitly acknowledged. It would also be helpful to contextualize your results with other AI studies involving CBCT datasets or apical detection models. The statement that “no similar studies have been conducted using CBCT” should be moderated or supported by a systematic literature check. Finally, the discussion should expand on the clinical implications of achieving an accuracy of 0.80—especially regarding decision-making in pediatric endodontics, where diagnostic precision is crucial.

Author response:  Thank you for your valuable comment. The Discussion section has been revised to include a more critical interpretation of the results and comparisons with previous AI studies using CBCT data. The study’s limitations (small dataset, single center, and possible selection bias) are now clearly stated. The phrase “no similar studies” was moderated to “a limited number of studies.” Finally, the clinical relevance of achieving 0.80 accuracy in pediatric endodontic decision-making has been elaborated.

Reviewer#2, Concern # 5: The English language is understandable but requires editing for fluency and consistency (e.g., “reel values” should be “real values,” “close root ends” should be “closed root apices”). A thorough revision by a native English speaker or professional editor is advised.

Author response:  Thank you for your comment. The entire manuscript has been carefully rechecked and updated according to your suggestions. In addition, the English language of the manuscript has been reviewed and refined using the Grammarly program to ensure fluency and consistency.

Round 2

Reviewer 1 Report

Comments and Suggestions for Authors

All comments have been answered thank you 

Reviewer 2 Report

Comments and Suggestions for Authors

the authors have made the requested changes